# Pediatric Sleep Questionnaire for Sleep Apnea in Newly Diagnosed Adolescent Idiopathic Scoliosis Patients

**DOI:** 10.3390/healthcare11182506

**Published:** 2023-09-10

**Authors:** Fatih Ugur, Kubra Topal, Mehmet Albayrak, Recep Taskin

**Affiliations:** 1Department of Orthopaedics and Traumatology, Kastamonu University Faculty of Medicine, Kastamonu 37150, Türkiye; rtaskin@kastamonu.edu.tr; 2Private Practice, Department of Otorhinolaryngology, Private Clinic, Kastamonu 37100, Türkiye; drkubratopal@gmail.com; 3Department of Orthopaedics and Traumatology, Ozel Tekirdag Yasam Hospital, Tekirdag 59020, Türkiye; doktorm.albayrak@gmail.com

**Keywords:** adolescent idiopathic scoliosis, obstructive sleep apnea, pediatric sleep questionnaire

## Abstract

Close association has been established between obstructive sleep apnea (OSA) and adolescent idiopathic scoliosis (AIS), with PSQ being employed as a screening method for OSA. A cross-sectional study was conducted among patients aged from 10 to 16 years who presented to a scoliosis outpatient clinic. Patient demographics, radiological assessments, and PSQ scores were gathered. A total of 299 patients were included in the study, with 28.7% males and 71.2% females. The average Cobb angle was 6.20°. PSQ scores revealed a prevalence of 33.4% for significant obstructive sleep apnea. Patients diagnosed with AIS exhibited a prevalence of 32.9% with positive PSQ results. Among those undergoing adenoid and/or tonsil surgery, 27% had positive PSQ scores. Factors such as genetics, abnormal biomechanical forces, environmental factors including melatonin, and intermittent hypoxia were explored for their potential contribution to AIS etiology. The aim of the study is to underscore the importance of early detection and intervention in OSA cases and highlights the effectiveness of the PSQ, as a screening tool in identifying sleep disorders. The findings underscore the complex relationship between OSA and AIS, and moreover any spinal curvature is in relation with OSA.

## 1. Introduction

Scoliosis is the most common cause of three-dimensional deformities of the spine and trunk characterized by a more than 10° curvature of the spine in the coronal plane [1,2]. The most common type of scoliosis is idiopathic scoliosis [3]. Overall idiopathic scoliosis prevalence is 0.47–5.2% worldwide in the current literature [1]. Recently, idiopathic scoliosis was classified into four subcategories according to age of onset: infantile, juvenile, adolescent, and adult [4]. Typically, adolescent idiopathic scoliosis (AIS) is more prevalent in females, and the disease tends to progress more than males [4,5]. Research across various disciplines including clinical medicine specialties, biology, genetic biomechanics, and anthropological studies has been conducted to investigate the etiology of AIS, a condition that progressively evolves and may impact quality of life in adulthood, with reported rates ranging from <1 to 3.7% [2,5]. Despite the name implying uncertainty, these studies have contributed to our understanding [1,2,3,4,5,6]. Idiopathic scoliosis is widely believed to result from multifactorial causes, with studies indicating that it does not stem solely from a single genetic factor, but rather involves a complex interplay of biological, mechanical, and hormonal factors contributing to its onset and progression [4,5].

The identification and prevention of factors driving idiopathic scoliosis are pivotal for potential improvement, heightening the importance in understanding risk and etiological factors [7].

Recent studies have been conducted regarding the effect of hypoxia on scoliosis [8,9,10]; the relationship between hypoxia and scoliosis was first demonstrated in congenital scoliosis, where curvature of the spine due to vertebral defects occurs in around 1 in 1000 live births. Studies have indicated that the genetic risk factor significantly amplifies the severity of vertebral defects in combination with brief gestational hypoxia, highlighting the impact of environmental factors [11,12]. Notably, morphological alterations have been observed in the intervertebral disc nucleus pulposus and chondrocytes, showcasing changes under hypoxic conditions [13,14,15].

In the study conducted by Mackintosh et al., it was discovered that 42.7% of scoliosis patients with symptoms such as snoring, apnea, and restless sleep were found to have sleep disorders [16]. Notably, 19.1% (13/68) of these cases underwent tonsil and/or adenoid removal for Obstructive Sleep Apnea (OSA).

Children with OSA experience partial or complete obstruction of the upper airway during sleep, leading to oxygen desaturation and hypercapnia, which is accompanied by sleep fragmentation [17,18]. Intermittent hypoxemia is now being acknowledged as a significant potential factor in the development of comorbidities related to OSA [18].

In addition to the hypoxia effects, sleep disruptions in children are recognized to negatively impact growth, behaviors, and cognitive functions, with inadequate and poor-quality sleep having well-established detrimental effects [16,19,20].

The gold standard for evaluating OSA and sleep disorders is polysomnogram (PSG). However, when considering neurobehavioral morbidity alongside sleep disorders, they prove to be insufficient. That is why the Pediatric Sleep Questionnaire (PSQ), developed and approved by Chervin and colleagues, specifically addresses respiratory-related sleep disorders [21,22,23].

Prior investigations have elucidated the interconnection among scoliosis, diminished oxygen supply meaning hypoxia, and sleep patterns. Nonetheless, except for the initial assessments of sleep disorders and OSA in scoliosis cases conducted by Mackintosh et al., comprehensive exploration in this realm has been scarce, and other evaluations of OSA have been lacking [11,12,13,14,15,16,20]. The significance of this relationship persists, insufficiently probed, with a specific dearth of knowledge regarding OSA prevalence. This study attempts to correct this gap by meticulously investigating the risk of OSA with PSQ through sleep assessments in patients who present with spinal curvature concern and are diagnosed with AIS.

## 2. Materials and Methods

The study was designed as a cross-section study. Ethical approval was obtained from the ethical board of our institution with the number of 2023-KEAK-62. The study was carried out in a single institution (Kastamonu Training and Research Hospital). Radiological and clinical data of children aged from 10 to 18, who sought care at the scoliosis outpatient clinic, were included [1].

Among the data collected from 322 patients, subsequent to comprehensive clinical, radiological, and physical examinations, it was ascertained that 3 patients were currently receiving epilepsy treatment, 4 patients exhibited congenital conditions, and the remaining cases presented either a familial predisposition to scoliosis or had previously undergone scoliosis treatment recommendations and follow-ups. Consequently, these 23 specific cases were excluded from the evaluation, resulting in a study cohort comprising a total of 299 participants.

All patients who presented to the scoliosis clinic either by themselves or referred from other clinics were included in the study, and their evaluations were conducted by a senior orthopedist.

Between June 2023 and August 2023, patients who presented with symptoms of scoliosis at the scoliosis clinic were included in the study if they had initially sought medical attention for the following complaints: curved posture, uneven shoulders, a more pronounced hump on one side of the back, a noticeable prominence while bending forward, rib deformities, cosmetic abnormalities, uneven breasts, and asymmetry in the chest wall [24].

Patients included in the study comprise not only those newly diagnosed with AIS, but also individuals with other spinal deformities that initially presented as nonstructural or functional scoliosis, including cases where radiographs reveal a spinal curvature due to asymmetric posture.

Exclusion criteria included participants with congenital, neurological, rheumatological, or cardiovascular diseases as well as those with psychotic disorders because of sleep disturbances. Patients previously diagnosed with scoliosis and treated or observed were not included [19].

Participants’ age, gender, and relevant demographic information were recorded.

In the clinical assessment, the primary objective is to uncover underlying causes and differentiate idiopathic cases through the history-taking phase. Factors such as family predisposition, existing health conditions, the presence of pain, and neurological symptoms, as well as shoulder and pelvic asymmetry and disparities in leg length are considered [25]. These findings are recorded for further analysis. Due to tonsillectomy +/− adenoidectomy being the predominant initial treatment for sleep-disordered breathing (SDB) [16,17,18] the presence of adenoid and tonsil hypertrophy and the need for surgery should be determined for patients. Systematic inquiry into the self-reported presence of adenoid and tonsil hypertrophy was carried out among the parents or guardians of the participants, as part of a comprehensive survey.

The evaluation was divided into two segments, both of which were conducted during the initial appointment. Participants, or their parents, meeting the criteria of this study provided informed consent by signing a consent form. They also completed questionnaires and underwent clinical assessments. The first part was scoliosis-specific clinical evaluations by the orthopedic surgeon, which included measurement of curve density with Cobb angle on a radiograph. The second part was the survey application.

Cobb angle measurements were conducted with a digital angle measurement tool. All clinicians were thoroughly trained in utilizing the Picture Archiving and Communication System database, executing digital Cobb angle measurements, and accurately documenting findings. Cobb angles, gauging coronal plane vertebral deformity on posteroanterior radiographs by measuring the angle between the superior endplate of a superior tilted vertebra and the inferior endplate of an inferior tilted vertebra, were assessed for each patient with vertebral curvatures [24,26]. Scoliosis is defined by a Cobb angle of 10° or more. It exhibits varying degrees of severity [1]. Mild (10–20°), moderate (20–40°), and severe (40° or above) forms are determined based on the Cobb angle [7]. The patients were divided into three groups according to the cobb angle, these are the patients with a cobb angle of 10° and above and the first time diagnosed with idiopathic scoliosis, those with a cobb angle of 5–9° and those with a cobb angle of 0–4°.

The validated PSQ is a validated screening instrument that aids in the diagnosis and management of various sleep-related breathing disorders encompassing OSA, spanning from snoring to severe presentations [21,22,23].

Sleep quality was evaluated using the Turkish version PSQ, with 22 questions answered by the parents of the patients that span four domains: sleep-related breathing, snoring, daytime somnolence, and behavior. Respondents indicate the presence or absence of each behavior or symptom by selecting “yes”, “no”, or “do not know”. Responses are scored as “0” (indicating the absence of the symptom) or “1” (indicating the presence of the symptom). PSQ score is calculated as the ratio between the sum of yes and number of answered questions.

The overall score is then calculated as a proportion of positive responses, with a score equal to or greater than 0.33 considered predictive for SDB. In children aged 2–18 years, the scale demonstrates strong internal consistency (α = 0.88) and exhibits a sensitivity of 0.85 along with a specificity of 0.87 for accurately identifying children with SRBD qualities [2]. The Turkish version of PSQ is also a reliable method for evaluating Turkish children with suggestive SDB symptoms [23].

### Statistical Analysis

The distribution of the data was examined using the Shapiro–Wilk test. In the comparison of three independent groups derived from subsets of patients categorized based on their Cobb angles, which did not follow a normal distribution, the Kruskal–Wallis test was applied. The statistical analysis was performed to understand whether there is a difference in PSQ among the three groups. When the marginal error is taken as 5% and the response distribution is taken as 33%, with a 95% confidence level and a significance level of 0.05, the sample size for 80% power is determined to be at least 291 individuals, with a prevalence of 5.2% [1]. For the evaluation of categorical data, the Fisher Freeman Halton test was employed. Numeric data were presented as median (min–max) and categorical data were presented as frequency (percentage). All statistical analyses were conducted and reported at a significance level of α = 0.05 using IBM SPSS Statistics 26.0 software.

## 3. Results

In this study the gender of the patients was classified as 86 (28.7%) male and 213 (71.2%) female. The average age was 11.54 ± 1.39 years (min: 10, max: 16). The average measured Cobb angle of the patients was determined to be 6.20 ± 5.15° (min: 0, max: 37).

When evaluating the reliability coefficient for the PSQ scale applied to the patients in our study, the Cronbach’s alpha reliability coefficient was 0.777, indicating good reliability. The mean total score obtained from the PSQ scale was 0.25 ± 0.17 (min: 0–max: 0.93). The distribution of responses to the questions in the demographic information form posed to the patients is summarized in Table 1.

The statistical analysis results regarding the evaluation of age, gender, and PSQ score value based on Cobb angle groups are presented in Table 2.

Upon examining Table 2, no statistically significant differences were found in terms of age, gender, and applied PSQ score among Cobb angle groups (*p* > 0.05). Although patients with higher Cobb angles (up to 37°) sought consultation at the scoliosis clinic, the analysis of PSQ results, focusing on positive scores, across all patients presenting due to lumbar curvature issues, did not yield any statistically significant differences within the defined Cobb angle groups.

The analysis results regarding the evaluation of the distribution of questions from the demographic information form posed to patients based on Cobb angle groups are provided in Table 3.

When Table 3 is examined, the distribution of demographic information questions among patients, based on Cobb angle groups, does not exhibit statistically significant differences (*p* > 0.05).

Out of the 299 patients who presented to the scoliosis clinic, 73 (24.4%) were diagnosed with AIS, with an average Cobb angle of 14° (ranging from 10 to 37°). Of the remaining 226 patients who applied, 89 (29.8%) had Cobb angles between 5 and 9°, while 137 (45.8%) exhibited no clinical or radiological pathology. Among patients diagnosed with AIS, 17 (23.3%) were male and 56 (76.7%) were female. Of them, 24 (32.9%) had PSQ scores ≥ 0.33, which were evaluated as significant results. The flowchart of the patients is seen in Figure 1.

For all the patients who applied to the scoliosis clinic, 100 (33.4%) had PSQ scores ≥ 0.33, which were considered significant results. Among these, 24 had Cobb angles greater than 10° (average 12.3°, ranging from 10 to 23°). Among the remaining 226 patients, 76 of them, which is 33.6%, had a positive PSQ test result (PSQ scores ≥ 0.33).

There was no statistical significance in terms of PSQ values for Cobb angles above 10°, and similarly, no statistically significant differences were observed for Cobb angles below 5°, in the range of 5–9°, and above 10°.

We found that 37 patients (12.3%) had undergone adenoid and/or tonsil surgery. Among them, 24 had undergone adenoidectomy, while the rest had undergone both tonsillectomy and adenoidectomy. Furthermore, 6 patients had Cobb angles greater than 10°, and only one of them had a significant PSQ value. Out of 37 patients, 10 were determined to be significant based solely on PSQ results. Regardless of the surgery, 48 patients were identified with tonsil hypertrophy (out of which 29 had no information about adenoid-related issues). Of the patients, 13 were identified with adenoid hypertrophy. In total, 93 patients (31.1%) had a history of adenoid and/or tonsil hypertrophy combined with a history of surgery.

The distribution of positive answers for each of the 22 individual questions in the PSQ is presented in Table 4. The order of individuals with the highest frequency of positive responses among these is presented in Table 5. Three out of the five questions with the highest affirmative responses are also present in the behavior (inattention domain) section of the questionnaire in Table 5.

## 4. Discussion

In this study, the risk of OSA was high, as expected considering the effect of hypoxia on scoliosis is reaching 32.9% among untreated patients diagnosed with AIS for the first time [8,9,10].

In the context of the typical human spinal structure, momentary and reversible curvatures towards either side manifest naturally in response to an uneven posture. Functionally speaking, such curvatures dissipate, and the spine returns to its straight alignment when the individual reclines or inclines sideways. Extended curvatures persist past a year in older children and usually become irreversible, even if the initial issue fades. Over time, ongoing uneven pressure can impact spinal growth plates. Yet, deformities might reverse if the posture is corrected and growth potential remains [7]. Hence, this elevated prevalence was not limited solely to AIS patients; it was also observed in patients with both Functional Curvature and Structural Curvature who attended the scoliosis outpatient clinic, with a prevalence of 33.4% [7].

Childhood OSA disrupts sleep due to upper airway problems, leading to cardiovascular, cognitive, and metabolic issues [17,18,27,28,29]. This can cause endothelial dysfunction, insulin resistance, metabolic problems, daytime sleepiness, and attention deficits [17]. Oxygen drops trigger inflammation and oxidative stress, alongside breathing struggles and chest pressure changes [17,18].

OSA, a prevalent condition in children, impacts 1–5% of the general pediatric population, typically manifesting between ages two and six [17,29]. OSA is the most severe condition on the spectrum of SDB. The PSQ is one of the most used and validated screening tools [30]. In comparison to Chervin et al.’s [21] initial validation and reliability study, Umano et al. [22] found that the PSQ displayed a remarkable 80% sensitivity and 100% specificity in detecting OSA, while the OSA-18 questionnaire, which is another questionnaire used for OSA diagnosis, showed a sensitivity of 94% and specificity of 55% for OSA diagnosis. These findings have also guided our application in patients, suggesting that in clinical settings with limited access to PSG, the PSQ could be utilized to classify disease severity and identify children in urgent need of sleep evaluation [22,27].

With data from 2010 suggesting a pediatric OSA prevalence rate of 4.8%, even though primary snoring has been reported at a prevalence of up to 15%, it is important to consider that 50% of even the highest reported prevalence may be attributed to OSA [31], since almost one-third of the patients in this study were found to be at high risk of OSA. Adenotonsillar hypertrophy is the main risk factor for OSA, treated primarily with adenotonsillectomy [28,32]. Recent studies show that adenoidectomy alone benefits non-obese patients with moderate OSA and small tonsils, reducing complications [32]. In the year 2010, a US study reported the adenotonsillectomy rate as 0.68%, whereas this rate was indicated to be 1.2% in a 2016 Swedish study [32]. In another study, based on data from 2017, the adenotonsillectomy prevalence was reported as 0.70% [33]. In this study, 20.3% of all patients were diagnosed with adenoid and/or tonsil hypertrophy, a similar percentage to the study reporting an adenoid hypertrophy rate of 19.8% and a tonsil hypertrophy rate of 16.2% in patients with SDB [23]. Furthermore, it was observed that 12.3% of these patients underwent adenoid and/or tonsillectomy. Furthermore, 32.3% of the patients have a history of adenoid tonsil hypertrophy or surgery.

MacKintosh et al. reported sleep abnormalities in EOS patients at a rate of 42.7% (68/159), and these patients included those with syndromic, neurological, and congenital etiologies; no idiopathic cases were included. Considering that most of these patients had Cobb angles exceeding 40° [16], it should be noted that this study, for the first time, revealed a high occurrence of sleep abnormalities and OSA not only in patients with idiopathic scoliosis but also in those presenting with scoliosis.

Yakut et al.’s [19] research on idiopathic scoliosis, utilizing the Pittsburgh Sleep Quality Index, revealed that a majority of participants (64.9%) indicated poor sleep quality. In their study regarding the pathogenesis of AIS, Girardo et al. [20] aimed to investigate the role of melatonin in human biology, specifically focusing on its involvement in circadian rhythm regulation and the management of sleep disturbances. Their research also encompassed an exploration of the clinical effectiveness and safety of melatonin across various pathological scenarios.

Animal experiment findings involving pinealectomy suggested that melatonin deficiency may play an important role in the development of experimental scoliosis. It is believed that this deficiency impairs the symmetrical growth of the proprioceptive system, which includes the paraspinal muscles and spine, and potentially contributes to the etiology of this condition [34]. Although the impact of disrupted melatonin secretion on scoliosis has been demonstrated in animal experiments, this effect has not been shown in humans. Studies of this nature underscore the necessity for conducting further research to enhance our understanding of the potential influence of multiple factors on the development of AIS.

Although there is a group of patients with uncertain AIS etiology, epidemiological studies strive to identify the causative agent. While the impact of genetic factors on scoliosis is evident in epidemiological research, the current evidence does not strongly support a sole genetic theory for the origin of AIS [1,2,3,4,5,6]. AIS’s exact causes remain elusive, but it is associated with genetic polymorphisms, disruptions in signaling peptides and hormones, and environmental triggers. While not all factors apply to every patient, various facets of these factors can be observed in each individual with AIS [35]. It is known that intermittent hypoxia (IH) affects not only numerous genes in various cell types but also through multiple pathways. IH-induced DNA methylation was tissue and cell selective. DNA methylation was also found in children with OSA who exhibited increased systemic inflammatory responses [36]. The study conducted by Meng and colleagues showed that increased curvature is linked to reduced methylation. These findings suggest that methylation at the specific site could potentially be a valuable biomarker for distinguishing between patients with and without curve progression in patients with AIS [37].

Between scoliosis and OSA, only a study conducted by Makintosh on EOS patients performed specifically in advanced cases has been found, and treatment has been initiated based on PSG results. No other study has been observed. Yakut et al.’s research on idiopathic scoliosis, using the Pittsburgh Sleep Quality Index, indicated that a majority of participants had sleep disturbances, unlike any other study in terms of sleep quality. However, there is no similar study for OSA. Considering this perspective, it can be seen that such a study is here conducted for the first time.

This study bears several limitations. First and foremost, the absence of clinical examinations for OSAS detracts from its comprehensiveness. The primary shortcoming lies in the lack of PSG assessments for all children, a gold standard for diagnosing SDB. Moreover, a notable gap in data exists concerning children with high body mass index, thus limiting the representation of a substantial proportion of those with severe obstructive sleep apnea. While there are existing studies on adolescent scoliosis, the study’s age range of 10–16 years introduces a constraint. Additionally, the research’s scope is restricted, excluding other types of scoliosis and narrowing the participant pool, further underscoring its limitations. Lastly, the cross-sectional design conducted within a single institution prevents an assessment of the patient population’s ethnic diversity, adding another layer to the study’s limitations.

## 5. Conclusions

This study emphasizes the intricate connection between OSA risk and scoliosis attitude as well as adolescent idiopathic scoliosis, underscoring the substantial risk posed by OSA in this cohort. The significant disruptive impact of OSA on sleep, coupled with its extensive cardiovascular, cognitive, and metabolic effects, underscores the urgency of early diagnosis and intervention. PSQ, especially in settings with limited access to PSG, is recognized as an effective screening tool that aids in assessing the severity of the condition. This study emphasizes the importance of evaluating spinal attitude when assessing the subject. It highlights the necessity to investigate hypoxia, particularly intermittent hypoxia when combined with OSA, as a risk factor for spinal attitude. This underscores the significance of exploring the relationship between these factors.

## Figures and Tables

**Figure 1 healthcare-11-02506-f001:**
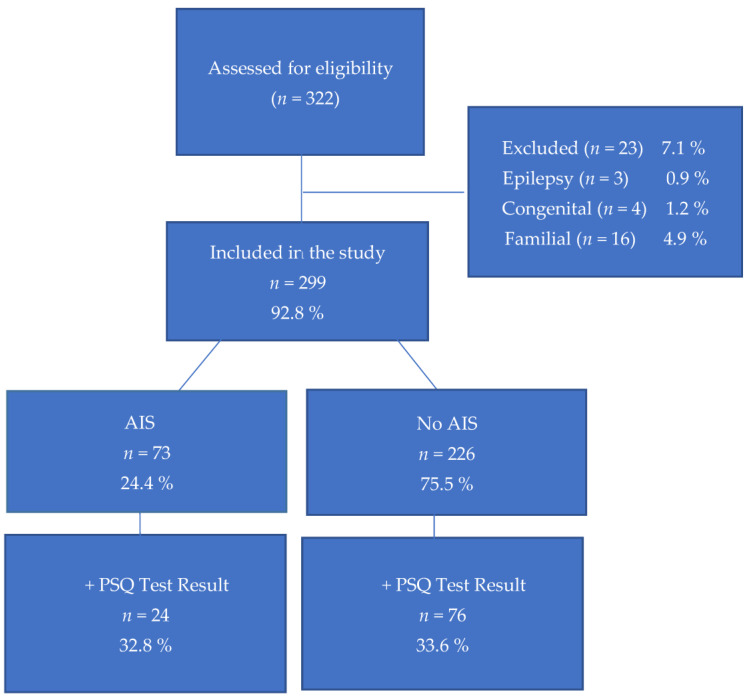
Flowchart of the patients.

**Table 1 healthcare-11-02506-t001:** Distribution of responses to the questions in the information form.

*n* = 299	Descriptive Statistics
**Is there, adenoid hypertrophy?**	
*Yes*	13 (4.3%)
*No*	217 (72.6%)
*Dont know*	69 (23%)
**Is there, tonsil hypertrophy?**	
*Yes*	48 (16%)
*No*	240 (80.3%)
*Dont know*	11 (3.6%)
**Has the patient undergone adenoid and tonsil surgery?**	
*Yes*	37 (12.3%)
*No*	260 (87.1%)
*Dont know*	2 (0.6%)
**PSQ score ≥ 0.33**	100 (33.4%)
	13 (4.3%)

The data are presented in terms of frequency (percentage).

**Table 2 healthcare-11-02506-t002:** Assessment of Age, Gender, and Total PSQ Score based on Cobb Angle Groups.

	Cobb < 5°	Cobb 5–9°	Cobb ≥ 10°	*p*-Value
**Age**	12 (10–14)	11.50 (10–15)	12 (10–16)	0.090 *
**PSQ total score**	0.22 (0–0.93)	0.23 (0–0.59)	0.20 (0–0.73)	0.980 *
**Gender**				
♂	40 (31%)	30 (30.2%)	16 (22.9%)	0.457 **
♀	89 (69%)	70 (69.8%)	54 (77.1%)	
**PSQ score**				
<0.33	99 (72.3%)	62 (69.7%)	49 (67.1%)	0.718 **
≥0.33	38 (27.7%)	27 (30.3%)	24 (32.9%)	

* *p*-value is for the Kruskal–Wallis test; ** *p*-value is for the Fisher–Freeman-Halton test; The data are presented as median (min-max) or frequency (percentage).

**Table 3 healthcare-11-02506-t003:** Evaluation of the distribution of questions from the demographic information form posed to patients based on Cobb angle groups.

	Cobb < 5°	Cobb 5–9°	Cobb ≥ 10°	*p*-Value
**Is there, adenoid hypertrophy?**				
*No*	97 (70.8%)	72 (80.9%)	51 (69.9%)	0.050 **
*Yes*	6 (4.4%)	2 (2.2%)	5 (6.8%)	
*Dont know*	34 (24.8%)	15 (16.9%)	17 (23.3%)	
**Is there, tonsil hypertrophy?**				
*No*	113 (82.5%)	70 (78.6%)	59 (80.8%)	0.941 **
*Yes*	21 (15.3%)	16 (17.9%)	11 (15.1%)	
*Dont know*	3 (2.2%)	3 (3.4%)	3 (2.2%)	
**Has the patient undergone adenoid and tonsil surgery?**				
*No*	118 (86.1%)	78 (87.6%)	66 (90.4%)	0.684 **
*Yes*	19 (13.9%)	12 (12.1%)	6 (8.2%)	
*Dont know*	1 (0.7%)	0 (0%)	1 (1.4%)	

** The *p*-value presented is for the Fisher—Freeman-Halton test. The data has been described in terms of frequency (percentage).

**Table 4 healthcare-11-02506-t004:** PSQ Questionnaire.

*Question*	*Total Positive Responses*
1: Snores more than half of the time	47
2: Always snores	8
3: Snores loudly	17
4: Has “heavy” or loud breathing	37
5: Has trouble breathing, or struggles to breathe	64
6: Seen your child stop breathing during the night	16
7: Tends to breathe through the mouth during the day	81
8: Has a dry mouth on waking up in the morning	172
9: Occasionally wets the bed	26
10: Wakes up feeling un-refreshed in the morning	162
11: Has problems with sleepiness during the day	114
12: Complains that they are sleepy during the day	75
13: It is hard to wake them up in the morning	115
14: Wakes up with headaches in the morning	38
15: Has stopped growing at a normal rate at any time since birth	30
16: Is overweight	58
17: Does not seem to listen when spoken to directly	97
18: Has difficulty organizing tasks and activities	91
19: Easily distracted by extraneous stimuli	140
20: Fidgets with hands or feet or squirms in seat	123
21: “On the go” or acts as if “driven by a motor”	62
22: Interrupts or intrudes on other (i.e., interferes with conversations/games)	95

**Table 5 healthcare-11-02506-t005:** Ranking of the questions with the most positive answers.

	*Question*	*Positive Responses*
1	Has a dry mouth on waking up in the morning	172
2	Wakes up feeling un-refreshed in the morning	162
3	Easily distracted by extraneous stimuli	140
4	Fidgets with hands or feet or squirms in seat	123
5	Interrupts or intrudes on other (i.e., interferes with conversation/games)	115

## Data Availability

The data sets used and/or analyzed during the present study are available from the first author on reasonable request.

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
