# Peer review of "Pediatric Sleep Questionnaire for Sleep Apnea in Newly Diagnosed Adolescent Idiopathic Scoliosis Patients"

_healthcare, 2023, doi:10.3390/healthcare11182506_

Round 1

Reviewer 1 Report

1. I think not many readers know abbreviation PSG (line 70) -must be clarified. Several times some abbreviations are explaned- one time is sufficient in the introduction.

2. Introduction can be more short.

3. Line 133- what it means "compromising 22 questions"?

4. First paragraph in Results must be in Materials and methods section.

5. One of the most important questions- why authors  have no control group, or at least provide data from literature about prevalence of OSA syndrome in so-called "healthy" adolescents. 

But authors very clearly explained all  several limitations of the study.

Author Response

Dear Reviewer 1, first of all, we would like to thank you for taking your time and making valuable contributions to our article. The article was reviewed and necessary revisions were made. We have mentioned your revisions as yellow in the text. I hope this reassures you. With our best regards.

R:Reviewer

A:Answer

1.R: I think not many readers know abbreviation PSG (line 70) -must be clarified. Several times some abbreviations are explaned- one time is sufficient in the introduction.

    A:PSG abbreviation was corrected as “polysomnogram”

2.R: Introduction can be more short.

    A:Introduction part was shortened

3.R:Line 133- what it means "compromising 22 questions"?

   A: ”compromising” deleted

  1. R:First paragraph in Results must be in Materials and methods section.

   A:The paragraph in the results section was moved to materials and methods section

  1. R:One of the most important questions- why authors  have no control group, or at least provide data from literature about prevalence of OSA syndrome in so-called "healthy" adolescents. 

   A:Data from literature about prevalence of OSA syndrome in so-called "healthy" adolescents was added.

For more details, please see the revised manuscript.

Reviewer 2 Report

This article is a prospective cohort study of adolescent idiopathic scoliosis (AIS) and obstructive sleep apnea (OSA). Although the number of cases is limited due to a single center, the design is well defined. The high complication rate suggests that there is an association between scoliosis and OSA, but this alone is not different from known previous reports. The unique features of this article are that AIS showed no difference in the association of OSA with other forms of scoliosis (L194-195) and that the severity of scoliosis in terms of Cobb angle was also not associated with OSA (table 2). These characteristics are important information for both otolaryngological and orthopedic treatment, and the Questionnaire assessment is not a weakness, as it will lead to further research on polygraphic contrasts.

 I would rate this study as good enough to be published if two things missing from this paper were corrected.

1) The first is the lack of discussion of power analysis for the association between AIS and OSA (L194-195) and the association between Cobb angle and OSA (table 2). (A software) G*power (https://www.psychologie.hhu.de/arbeitsgruppen/allgemeine- psychologie-und-arbeitspsychologie/gpower).

2) The second point is that you are trying to explain the association between OSA and melatonin with scoliosis (L268-274), but in table 3, please discuss why only adenoid is associated with the Cobb angle. If adenoid is clearly more severe than tonsil hypertrophy, that would make sense, but if you have such a reference, please cite it. The results of this study do not explain why table 2 shows no association between Cobb angle and PSQ score.

Please correct the above two points.

Author Response

Dear Reviewer 2, first of all, we would like to thank you for taking your time and making valuable contributions to our article. The article was reviewed and necessary revisions were made. We have mentioned your revisions as red in the text. I hope this reassures you. With our best regards.

R:Reviewer

A:Answer

  1. R: The first is the lack of discussion of power analysis for the association between AIS and OSA (L194-195) and the association between Cobb angle and OSA (table 2). (A software) G*power (https://www.psychologie.hhu.de/arbeitsgruppen/allgemeine- psychologie-und-arbeitspsychologie/gpower).

    A: power analysis was calculated and added in statistical analysis section

  1. R:The second point is that you are trying to explain the association between OSA and melatonin with scoliosis (L268-274), but in table 3, please discuss why only adenoid is associated with the Cobb angle. If adenoid is clearly more severe than tonsil hypertrophy, that would make sense, but if you have such a reference, please cite it. The results of this study do not explain why table 2 shows no association between Cobb angle and PSQ score.

   A: In our study, our primary intention was to underscore the influence of sleep on the relationship between scoliosis. It is essential to note that our aim was not to establish a definitive link between melatonin and scoliosis, as indicated by the absence of conclusive data in the existing discussions (as highlighted in the statement: "numerous studies exploring the impact of melatonin, a hormone released during sleep"). Taking your observation into account, we have revised the abstract as follows: "Factors such as genetics, abnormal biomechanical forces, environmental factors including melatonin, and intermittent hypoxia were explored for their potential contribution to AIS etiology," in accordance with your guidance and the relevant references. Regarding the differentiation between tonsils and adenoids, we have revised Table 3 to ensure clarity. Both adenoid hypertrophy and tonsil hypertrophy are now listed as separate rows in the table. Furthermore, the query "Has the patient undergone adenoid and tonsil surgery?" has been placed in a distinct row for clarity. We want to emphasize that our study extensively investigated both adenoid and tonsil hypertrophy, gathering information through surgical procedures. Out of the total of 37 patients, 24 underwent adenoidectomy, while 13 patients underwent both tonsil and adenoid surgeries. As you astutely observed, adenoid surgery is more prevalent. However, it is important to note that there was no statistical correlation established with the Cobb angle. Notably, in support of this observation, previous research (referenced as (33)) highlights the efficacy of adenoidectomy specifically in treating OSA in cases of tonsil reduction.

For more details, please see the revised manuscript.

Reviewer 3 Report

Comments to the authors

Thank you for the invitation to review the manuscript entitled “Pediatric sleep questionnaire for sleep apnea in newly diagnosed adolescent idiopathic scoliosis patients”. This is a cross-sectional study that look at the prevalence of OSA among adolescents diagnosed with AIS.

A Major concern with this paper is that the authors claim to be able to reach an assessment of OSA prevalence through the PSQ, while the PSQ is only a screening method and not a diagnostic tool (see below my extensive comments). Also, some of the sections were confused and did not follow a nice flow. Statistica analysis is lacking and can be further explored. Abstract overcomes the word limit as allowed by the author guidelines. Discussion lacks a re-elaboration and speculation of the finginds. Conclusions do not present the findings of the study.

Abstract: the abstract lacks the explication of the study aim, which is important to be highlighted. Moreover, in the method section, the reader may not know what the Cobb angle is, which is worthy a further detail. I urge the authors to be careful with their statement of “PSQ scores revealed a prevalence of 33.4% for significant OSA”. The PSQ is only a screening tool, and this should be reflected in this statement to not be misleading.

The authors should indicate what they consider as “positive results” of PSQ. Did they put a cut-off? Finally, the accepted guidelines of the journal indicate a maximum of 200 words for the abstract. The current version is 317 words. The authors need to cut the words down to comply with the guidelines.

Minor edit: replace “obstructive sleep apnea” with “OSA”, as it has already been used as acronym (p 1, line 22)

Introduction: this paragraph is confused and needs to be re-organized. It does not follow a structure. See below my comments and clarification

-        Pag 1, line 40: are these prevalence figures for overall idiopathic scoliosis and worldwide data? This should be clarified

-        Page 1, line 45: it is not clear if this sentence refers to AIS or to idiopathic scoliosis in general. I guess that it is referred to AIS, considering that in page 2 line 47 other estimates of prevalence are provided. If it is AIS, please use the acronym AIS instead of idiopathic scoliosis

-        Page 2, lines 45-49. This is not clear. It is stated that we do have a better understanding of the etiopathogenesis of idiopathic scoliosis, although it is not solely conducted to one gene. So what is our understanding? What are the other factors implied in the etiopathogenesis?

-        Page 2, line 54. It is not clear where the “congenital scoliosis” comes from. This has to be introduced earlier in the introduction so that the reader understands why hypoxia may be important in the congenital scoliosis, to begin with, and then also understands why it has been brought up within the idiopathic scoliosis.

-        Where does the EOS come from? In page 1 this is not mentioned among the possible classification of scoliosis. It needs to be either introduced earlier, or removed.

-        The paragraph on OSA is confused. Definition is first provided, then consequences of poor sleep quality. Yet, then the authors mentioned the use of melatonin for idiopathic scoliosis (and are we talking about AIS? Or general idiopathic scoliosis?); then they moved back to talk about the screening of sleep abnormalities (why not only OSA?); and then diagnosis with PSG. The parapraph needs to follow a better flow (e.g, remove the mention of melatonin, or move it once the probable association between scoliosis and OSA is further explored)

-        Then I encourage either to rephrase or to quote a reference to the page 2, line 70-71. The diagnosis of OSA in children is with PSG. The use of other forms such as cognitive and behavioral disorders may help with the screening of high risk patients, but they are far from being diagnostic tools.

-        Page 2, line 81-83: PSG is a screening method. It can elucidate about the RISK, not about the OSA prevalence (if it is not added to a PSG). Also, the aim of the study needs to clarify that this will be investigated within the adolescents presenting with AIS. There is some missing link with the fact that the authors are going to investigate OSA prevalence within adolescents; however, the introduction is all based on children, which the reader would expect that not AIS but IS in children to be evaluated. The introduction needs to be modified to reflect that the paper will be based on AIS on adolescents

Methods:

-        Line 88-89: any max limit of age within the included children?

-        Were patients with previous tonsillectomy or adenoidectomy excluded from the study?

-       Paragraph of Cobb angles (from lines 120 to 128): the authors need to indicate which measures of Cobb angles are pathological and which are the normal values. Does the angle identify different degree of severity of AIS? This is worth mentioning

-       Line 130-131: how can the PSQ be a tool for treatment of OSA? Maybe for monitoring the treatment, but the PSQ does not treat OSA (nor diagnose)

-        Line 143-144: was the Turkish version utilized in the current study? This is not clear

-       No sample size calculation was performed nor power analysis

-        Who answered the PSQ? The parents or the patient?

Statistical analysis: which independent groups? The ones derived from the Cobb angles? Need to be specified.

-      What is the purpose of the statistical analysis? Understand whether there is a difference in PSQ among the three groups? This has not been stated.

-      I suggest that the authors can add a logistical analysis with continuous data from Cobb angle and data from the PSQ to see whether there is an association

-     Was there any difference in the individual item frequency of PSQ among the three groups?

-     The authors should also compare the frequency of high risk of OSA between group IAS vs non-AIS

Results: the results are confused in that it is difficult to understand if the authors are presenting the data from the total sample or from those diagnosed wit AIS (e.g. lines 200-207)

-       It is not clear where the outcomes investigated in table 1 and table 3 derived from. Nothing makes the reader think that the degree of adenoid / tonsil hypertrophy was exmined. Was this part of the clinical examination? If that is the case, how were adenoid and tonsils evaluated? Following which validated scale?

-       The reference to Figure 1 is not embedded within the text. Also, I suggest that the authors add the % beside the numbers, so that it is clearer to th reader what is the extent of each group

Discussion: the first statement is misleading (and also line 247-8), as this study does not have the power to assess prevalence of OSA. Again, this study only assesses the risk, the prediction of OSA . Moreover, this is not the findings of the study. The findings of the study are that there is no difference between the 3 groups.

-       Line 265 is unclear. Which diverse population do the authors refer to?

-       The discussion is lacking in the sense that it does not speculate nor re-elaborate on the findings of the study

Limitation: the BMI was not taken into consideration, as contributing factors to both conditions

Conclusion: these do not reflect the current findings nor they restate them. This Needs to be addressed.

Minor changes are needed 

Author Response

Thank you for your valuable contribution and positive comments. Upon your request, the article has been revised and necessary corrections have been made. We have mentioned yor revisions as green in the text. With all due respect, we hope our revision has satisfied you.

R:Reviewer

A:Answer

Abstract

R:the abstract lacks the explication of the study aim, which is important to be highlighted. Moreover, in the method section, the reader may not know what the Cobb angle is, which is worthy a further detail. I urge the authors to be careful with their statement of “PSQ scores revealed a prevalence of 33.4% for significant OSA”. The PSQ is only a screening tool, and this should be reflected in this statement to not be misleading.

A: The aim is highlighted, Cobb angle was removed from the abstract,PSQ is a screening tool is highlighted

R:The authors should indicate what they consider as “positive results” of PSQ. Did they put a cut-off? Finally, the accepted guidelines of the journal indicate a maximum of 200 words for the abstract. The current version is 317 words. The authors need to cut the words down to comply with the guidelines.

A: Positive results was removed from the article, word count was decreased as wanted

R:Minor edit: replace “obstructive sleep apnea” with “OSA”, as it has already been used as acronym (p 1, line 22)

A: OSA was mentioned as an acronym

Introduction

R:Page 1, line 40: are these prevalence figures for overall idiopathic scoliosis and worldwide data? This should be clarified

A: The referenced study with the number [1] titled "Epidemiology of Adolescent Idiopathic Scoliosis" presents prevalence results from two distinct studies. Kamtsiuris et al reported a prevalence of 5.2% based on their study involving 17,641 children, while Cilli et al found a prevalence of 0.47% in their study involving 3,175 children. These findings, highlighted in the referenced table, are extracted from a total of seven studies encompassing a cumulative child population of 1,423,390. Mentioned in the text

R: Page 1, line 45: it is not clear if this sentence refers to AIS or to idiopathic scoliosis in general. I guess that it is referred to AIS, considering that in page 2 line 47 other estimates of prevalence are provided. If it is AIS, please use the acronym AIS instead of idiopathic scoliosis

A: AIS was used as an acronym

R: Page 2, lines 45-49. This is not clear. It is stated that we do have a better understanding of the etiopathogenesis of idiopathic scoliosis, although it is not solely conducted to one gene. So what is our understanding? What are the other factors implied in the etiopathogenesis?

A: The inclusion of other relevant factors has enriched the content and brought the study in alignment with the references provided.

R:      Page 2, line 54. It is not clear where the “congenital scoliosis” comes from. This has to be introduced earlier in the introduction so that the reader understands why hypoxia may be important in the congenital scoliosis, to begin with, and then also understands why it has been brought up within the idiopathic scoliosis.

A: Corrections and additions were made in accordance with the suggestions

R:Where does the EOS come from? In page 1 this is not mentioned among the possible classification of scoliosis. It needs to be either introduced earlier, or removed.

A: EOS was removed

R:The paragraph on OSA is confused. Definition is first provided, then consequences of poor sleep quality. Yet, then the authors mentioned the use of melatonin for idiopathic scoliosis (and are we talking about AIS? Or general idiopathic scoliosis?); then they moved back to talk about the screening of sleep abnormalities (why not only OSA?); and then diagnosis with PSG. The parapraph needs to follow a better flow (e.g, remove the mention of melatonin, or move it once the probable association between scoliosis and OSA is further explored)

A: Melatonin was removed and the sentence was revised.

R:Then I encourage either to rephrase or to quote a reference to the page 2, line 70-71.

A: It was rephrased again. Charvin and colleagues (21), for the first time, utilized the PSQ in the prediction of sleep apnea. They pointed out in their own words that the insufficiency of PSG in predicting neurobehavioral morbidity makes this test more effective. The term "neurobehavioral morbidity" was similarly expressed by Yuksel and colleagues (23).

R:The diagnosis of OSA in children is with PSG. The use of other forms such as cognitive and behavioral disorders may help with the screening of high risk patients, but they are far from being diagnostic tools.

A: In accordance with your suggestion, the sentence has been revised, replacing "cognitive and behavioral disorders" with "neurobehavioral morbidity.This study attempts to correct this gap by meticulously investigating the risk of OSA with PSQ through sleep assessments in patients who present with spinal curvature concern and are diagnosed with AIS.”

R: Page 2, line 81-83: PSG is a screening method. It can elucidate about the RISK, not about the OSA prevalence (if it is not added to a PSG). Also, the aim of the study needs to clarify that this will be investigated within the adolescents presenting with AIS. There is some missing link with the fact that the authors are going to investigate OSA prevalence within adolescents; however, the introduction is all based on children, which the reader would expect that not AIS but IS in children to be evaluated. The introduction needs to be modified to reflect that the paper will be based on AIS on adolescents

A:The introduction was modified to reflect that the paper will be based on AIS on adolescents, the risk was added.

Methods

R:Line 88-89: any max limit of age within the included children?

A:The age range was corrected to 10-18 years, and a reference was added.

R:Were patients with previous tonsillectomy or adenoidectomy excluded from the study?

A:No they are not excluded the surgical history of the participants was documented. The collected data, including whether they underwent surgery or not, was then utilized for statistical analysis.

R:Paragraph of Cobb angles (from lines 120 to 128): the authors need to indicate which measures of Cobb angles are pathological and which are the normal values. Does the angle identify different degree of severity of AIS? This is worth mentioning

A:The Cobb angle was defined, and the severity of the condition was indicated. Furthermore, a reference was incorporated

R:Line 130-131: how can the PSQ be a tool for treatment of OSA? Maybe for monitoring the treatment, but the PSQ does not treat OSA (nor diagnose)

A:Corrected

R:Line 143-144: was the Turkish version utilized in the current study? This is not clear

A:”Turkish version” is clarified and was added

R:No sample size calculation was performed nor power analysis

A:both were calculated and added in statistical analysis section. Rabelled as red

R:Who answered the PSQ? The parents or the patient?

A:Parents answered (mentioned in manuscript)

Statistical Analysis

R:which independent groups? The ones derived from the Cobb angles? Need to be specified.

A:The independent groups mentioned in the study refer to the subsets of patients categorized based on their Cobb angles.

R:What is the purpose of the statistical analysis? Understand whether there is a difference in PSQ among the three groups? This has not been stated.

A:no statistically significant differences were found in terms of age, gender, and applied PSQ score among Cobb angle groups (p > 0.05). This was mentioned in Table 2.

R: I suggest that the authors can add a logistical analysis with continuous data from Cobb angle and data from the PSQ to see whether there is an association

A:There was no statistical significance in terms of PSQ values for Cobb angles above 10 degrees, and similarly, no statistically significant differences were observed for Cobb angles below 5 degrees, in the range of 5-9 degrees, and above 10 degrees. Mentioned in results part

R:Was there any difference in the individual item frequency of PSQ among the three groups?

A:Table 2, no statistically significant differences were found in terms of age, gender, and applied PSQ score among Cobb angle groups (p > 0.05). mentioned in the results part

R:The authors should also compare the frequency of high risk of OSA between group IAS vs non-AIS

A:The attempt was made to establish this by comparing the frequency of the group with Cobb angles of 10 degrees and above with the frequencies of the other groups according to Cobb angles. Mentioned in the results section

Results

R:the results are confused in that it is difficult to understand if the authors are presenting the data from the total sample or from those diagnosed wit AIS (e.g. lines 200-207)

A:Following your guidance, Figure 1 has been updated to include information within this paragraph. The sentences have been revised to eliminate confusion and address the concerns raised in your feedback.

R:It is not clear where the outcomes investigated in table 1 and table 3 derived from. Nothing makes the reader think that the degree of adenoid / tonsil hypertrophy was exmined. Was this part of the clinical examination? If that is the case, how were adenoid and tonsils evaluated? Following which validated scale?

A:The most important surgical approach in OSA treatment is well-known to be the removal of tonsils and/or adenoids. Therefore, when investigating OSA, considering adenoid and tonsil hypertrophy is crucial. Hence, along with a questionnaire, families were also asked to answer these questions, as the ranking solely reflects the seriousness they attach to this condition.

Mentioned in the manuscript

R:The reference to Figure 1 is not embedded within the text. Also, I suggest that the authors add the % beside the numbers, so that it is clearer to th reader what is the extent of each group

A:Reference to Figure 1 is embedded in the text, and the percentages were added.

Discussion

R:the first statement is misleading (and also line 247-8), as this study does not have the power to assess prevalence of OSA. Again, this study only assesses the risk, the prediction of OSA . Moreover, this is not the findings of the study. The findings of the study are that there is no difference between the 3 groups.

A:was corrected as risk of OSA

R:Line 265 is unclear. Which diverse population do the authors refer to?

A:was removed

R:The discussion is lacking in the sense that it does not speculate nor re-elaborate on the findings of the study

A:the findings of the study was compared and mentioned in the discussion section

Limitations

R:the BMI was not taken into consideration, as contributing factors to both conditions

A: BMI was mentioned

R:these do not reflect the current findings nor they restate them. This Needs to be addressed.

A:Conclusion part was revised

For more details, please see the revised manuscript.

Round 2

Reviewer 3 Report

The authors have done a good job in addressing my previous concerns. 

Some minor points still need to be tackled. 

Abstract: the abstract should be a stand-alone description. Therefore, as you needed to cut the words down, you should cut the introduction and better summarize the methods, instead of removing the results. Please, reformulate the abstract following my recommendations. (e.g., remove the red highlights in red)

Lines 117-118: English needs to be polished and improved here. Something along the line of "Self-reported presence of adenoid and tonsil hypertrophy was investigated among participant's parents or guardian" 

Lines 138: if you have to briefly define PSQ, I would never define it as "economical instrument". It is a validated screening instrument, and not used for management of sleep-related breathing disorders. This is not even mentioned in the three references that the authors provided. Please, correct it this misleading information.

Lines 159-162: this should be supported by a reference 

English language may require a revision. 

Author Response

Dear Reviewer 3

Once again thank you for your valuable contribution and positive comments. Upon your request, the article has been revised and necessary corrections have been made. We have mentioned yor revisions as green in the text. With all due respect, we hope our revision has satisfied you.
